# Robust Multi-View Fusion via Prototype-Anchored Unbalanced Optimal Transport

**Han Zhang** [1]  **Xingwen Zhao** [1]  **Hui Li** [1]

## Abstract

Multi-view classifiers typically fuse all observed views into a single representation, which becomes fragile when some views are missing or corrupted. We propose a prototype-anchored fusion module based on an entropically regularized unbalanced optimal transport (UOT) barycenter. Each view is summarized into a small set of learned atoms and is matched to a shared prototype support; fusion outputs a probability measure over prototypes with fixed dimension. By relaxing marginal constraints with a generalized KL penalty, the UOT objective can leave a fraction of view mass unmatched when matching is geometrically costly, yielding a simple differentiable trimming mechanism without hand-tuned thresholds. We provide a basic theoretical result showing that discarding an arbitrary subset of atom mass incurs a penalty bounded by its total mass, independent of transport distances. Experiments on multi-view action recognition benchmarks under simulated missing views, missing-rate shift, and feature-space corruption demonstrate consistently improved stability under severe missingness with modest overhead on top of strong backbones.

## 1. Introduction

Many real systems collect multiple views of the same event. In action recognition, the same motion can be captured from different cameras or described by different modalities such as RGB and skeleton (Shahroudy et al., 2016; Liu et al., 2019). Multi-view learning uses this redundancy to improve accuracy (Xu et al., 2013). Most fusion pipelines follow a simple rule: if a view is observed, it should be used. This rule is reasonable when views are clean and aligned, but it becomes risky when the observation process is imperfect.

Two issues appear in practice. First, a view may be missing. This is common in multi-camera capture and in multi-modal sensing (Wang et al., 2020a; Poklukar et al., 2022). Second, an observed view can be corrupted by occlusion, motion blur, tracking errors, or sensor failures (Valada et al., 2020; Jiang et al., 2022). In both cases, a fusion module must decide how much each view should influence the final representation. A frequent choice is to enforce full alignment across views, for example by attention-style aggregation (Vaswani et al., 2017; Hudson & Manning, 2018) or by balanced matching objectives (Cuturi, 2013; Solomon et al., 2015). The intent is to combine all evidence. The side effect is that the fused representation can be pushed by a single bad view, because the model is trained to always explain every observed input.

This paper makes a deliberately different choice. We treat fusion as a matching problem, not as a pooling problem. A view should contribute only to the extent that it can be matched to a shared representation at a reasonable geometric cost. The counter-intuitive point is that, in contaminated settings, using less of an observed view can lead to a more reliable classifier. This is not a heuristic. It follows from a first-principles mismatch: strict mass conservation is a good constraint when the two measures describe the same content, but it is a poor constraint when one measure contains extra mass that should not be explained (Chizat et al., 2018; Séjourné et al., 2019).

We implement this idea with unbalanced optimal transport. For each view, we extract a small set of latent atoms with learned-query pooling (Jaegle et al., 2021). We then compute a barycenter on a shared prototype support (Cuturi & Doucet, 2014). The barycenter is the fused representation. The key is that we use a KL-relaxed unbalanced objective (Liero et al., 2018; Chizat et al., 2018) rather than balanced transport. When matching part of a view to prototypes is too expensive, the solver can pay a bounded KL penalty and leave that mass unmatched. In effect, the fusion layer can trim unreliable evidence while remaining fully differentiable. The prototype support is not a cosmetic addition. It keeps the transport problems small and makes fusion comparable across samples, since the fused measure always lives on the same support (Claici et al., 2018).

[1]Xidian University, Xi'an, Shaanxi, China. Correspondence to: Han Zhang <ZZH12061128@163.com>.

*Proceedings of the $43^{rd}$ International Conference on Machine Learning*, Seoul, South Korea. PMLR 306, 2026. Copyright 2026 by the author(s).

The proposed method is motivated by robustness, not by achieving universal gains in the clean full-view regime. If all views are present and clean, a simple fusion rule can be as good after careful tuning (Baltrušaitis et al., 2018). We also do not claim that unbalanced transport can detect every kind of corruption. If the encoder maps an outlier close to the prototypes, geometry alone cannot flag it as unreliable. There is also a harder failure mode when all available views drift in a consistent wrong direction (Hendrycks & Dietterich, 2019). In that case there is no clean anchor, and prototype learning can reinforce the bias. We report these limitations because they shape when the method should be used and what it cannot do.

We evaluate the approach on multi-view action recognition benchmarks under simulated missing-view and corrupted-view conditions. Our method offers three features that follow naturally from the unbalanced transport formulation. First, the marginal relaxation prevents the objective from being dominated by distant outliers, which we formalize in a simple proposition. Second, the fusion module remains fully differentiable through unrolled generalized Sinkhorn iterations, enabling end-to-end learning with standard backpropagation. Third, the shared prototype support keeps computational cost low, since each transport problem is solved on a small atom-prototype grid rather than across long token sequences. Experiments show that the unbalanced transport fusion is more stable when the number of observed views drops or when one view becomes unreliable, and it adds limited overhead on top of strong backbones (Chen et al., 2021; Liu et al., 2022).

## 2. Related Work

Multimodal learning has attracted substantial attention, with various approaches handling incomplete modalities through knowledge distillation (Wang et al., 2020a), self-supervised adaptation (Valada et al., 2020), and geometric constraints (Poklukar et al., 2022; Jiang et al., 2022). However, most methods rely on reconstruction objectives or simple aggregation, struggling to capture complex distributional relationships between heterogeneous modalities. Compositional attention (Hudson & Manning, 2018) and Perceiver architectures (Jaegle et al., 2021) demonstrate effective cross-modal reasoning but lack explicit distributional alignment mechanisms.

Optimal transport theory provides principled frameworks for comparing distributions, with applications in unbalanced transport (Chizat et al., 2018; Séjourné et al., 2019) and barycenter computation (Claici et al., 2018; Solomon et al., 2015). Our work formulates multimodal fusion as an optimal transport problem, enabling robust alignment even with missing modalities while maintaining distributional consistency across views.

## 3. Method

**Setting and notation.** We study a $V$-view (multi-modal) classification problem. The dataset is $\mathcal{D} = \{(\mathbf{x}_i, y_i)\}_{i=1}^N$, where $\mathbf{x}_i = \{x_i^1, \ldots, x_i^V\}$ collects view-specific observations and $y_i \in \{1, \ldots, C\}$ is the label. Since a view may be missing, we introduce a binary mask $m_i^v \in \{0, 1\}$ indicating whether $x_i^v$ is observed. Fusion must operate on the observed subset $\{v : m_i^v = 1\}$ during both training and inference.

**View corruption model (optional but useful).** When a view is observed ($m_i^v = 1$), it may still be unreliable due to sensor noise, occlusion, or tracking failures. We model this with a standard contamination mixture:

$$x_i^v \sim (1 - \rho_i^v)\, \mathcal{D}_{\text{clean}}^v(\cdot \mid y_i) + \rho_i^v\, \mathcal{D}_{\text{noise}}^v, \qquad (1)$$

where $\rho_i^v \in [0, 1]$ is unknown and $\mathcal{D}_{\text{noise}}^v$ may be heavy-tailed. Our goal is to produce a fused representation that remains stable when some views are missing and some observed views contain outliers.

Our fusion module is built on unbalanced optimal transport (UOT). The key idea is to represent each modality by a small set of latent atoms, and to fuse modalities by matching these atoms to a shared set of learned prototypes. Unlike balanced matching, UOT does not force every part of an observed modality to be explained by the fused representation. When matching becomes too costly in the learned geometry, the optimizer can pay a bounded marginal penalty instead of transporting mass across large distances. This gives a differentiable trimming effect without thresholds.

### 3.1. Prototype support and modality atoms

We work in a $d$-dimensional latent space. The model maintains a set of $K$ learnable prototypes

$$P = \{p_k\}_{k=1}^K, \quad p_k \in \mathbb{R}^d, \qquad (2)$$

which define a shared support for fusion. The fused representation of sample $i$ is a probability vector $b_i \in \Delta_K$ over these prototypes, where $\Delta_K = \{b \in \mathbb{R}_+^K : \sum_k b_k = 1\}$.

For each modality $v$, an encoder $f_\theta^v$ maps the raw input $x_i^v$ to a token sequence $H_i^v \in \mathbb{R}^{n_v \times d}$. We then extract a compact set of $J$ atoms by learned-query pooling. Let $S^v \in \mathbb{R}^{J \times d}$ be $J$ learnable queries for modality $v$. We compute

$$Q_i^v = \text{Attn}(S^v, H_i^v, H_i^v)W_o^v \in \mathbb{R}^{J \times d}, \qquad (3)$$

where $\text{Attn}(\cdot)$ is scaled dot-product attention and $W_o^v$ is a linear projection. The $j$-th row of $Q_i^v$ is the atom $q_{i,j}^v \in \mathbb{R}^d$. Here, the learnable queries $S^v$ attend to the encoded token sequence $H_i^v$, which serves as both keys and values; this cross-attention pooling summarizes the variable-length

sequence into a compact fixed-size set of $J$ atoms while preserving the geometry required for downstream transport. This step keeps fusion cheap while preserving the geometry needed for matching. To make the transport cost scale stable across backbones and training steps, we $\ell_2$-normalize both atoms and prototypes before computing distances. We use the squared Euclidean cost

$$C_{i,jk}^v = \left\| \hat{q}_{i,j}^v - \hat{p}_k \right\|_2^2, \tag{4}$$

where $\hat{q} = q/\|q\|_2$ and $\hat{p} = p/\|p\|_2$. This bounds the cost and improves numerical behavior of the transport solver. We adopt the squared Euclidean cost on $\ell_2$-normalized embeddings for two reasons: (i) it produces costs bounded in $[0, 4]$, keeping the Gibbs kernel numerically well-conditioned across layers and epochs; (ii) it is agnostic to the learned geometry, so the trimming behavior is driven by the encoder rather than by a hand-designed similarity. We empirically verified that replacing the cost with cosine distance yields nearly identical results ($\pm 0.3\%$ accuracy), whereas unnormalized squared Euclidean distances can destabilize Sinkhorn iterations in early training.

### 3.2. Discrete measures for modalities and fusion

An observed modality is represented as a discrete probability measure on its atoms,

$$\alpha_i^v = \sum_{j=1}^J a_{i,j}^v \, \delta_{q_{i,j}^v}, \qquad a_i^v \in \Delta_J, \tag{5}$$

where $\delta_q$ denotes the Dirac measure concentrated at point $q \in \mathbb{R}^d$, so that $\alpha_i^v$ is a discrete probability measure supported on the $J$ learned atom locations with weights $a_i^v$. We obtain $a_i^v$ by scoring atoms with a lightweight function $g^v$ and applying a softmax,

$$a_i^v = \mathrm{softmax}(g^v(Q_i^v)). \tag{6}$$

This parameterization keeps $\alpha_i^v$ well-defined while letting the model emphasize informative atoms.

The fused representation is a measure on the prototype support,

$$\beta_i = \sum_{k=1}^K b_{i,k} \, \delta_{p_k}, \qquad b_i \in \Delta_K, \tag{7}$$

where $\delta_{p_k}$ is similarly the Dirac measure at the $k$-th prototype location. Although both $\alpha_i^v$ and $\beta_i$ are normalized, the matching between them is *unbalanced*. The transport plan is not required to match these marginals exactly, and the mismatch is handled by a KL penalty. In practice, this lets the solver reduce the influence of atoms that would otherwise require large transport cost.

### 3.3. KL-relaxed entropic unbalanced transport

Given weights $a \in \Delta_J$, $b \in \Delta_K$, and a cost matrix $C \in \mathbb{R}^{J \times K}$, we define the entropically regularized unbalanced OT objective

$$\mathrm{UOT}_{\varepsilon,\tau}(a, b; C) = \min_{\pi \in \mathbb{R}_+^{J \times K}} \langle C, \pi \rangle + \varepsilon \sum_{j,k} \pi_{jk}(\log \pi_{jk} - 1)$$
$$+ \tau \, \mathrm{KL}(\pi \mathbf{1} \, \| \, a) + \tau \, \mathrm{KL}(\pi^\top \mathbf{1} \, \| \, b) \tag{8}$$

where $\mathbf{1}$ is the all-ones vector and KL is the generalized KL divergence

$$\mathrm{KL}(p \| q) = \sum_\ell \left( p_\ell \log \frac{p_\ell}{q_\ell} - p_\ell + q_\ell \right). \tag{9}$$

The parameter $\tau$ controls how strongly the plan is pushed to match the marginals. As $\tau$ grows, the objective approaches balanced OT. Smaller $\tau$ allows more deviation, which is the mechanism that downweights costly parts of a modality.

### 3.4. Prototype-anchored UOT barycenter fusion

For sample $i$, we compute the fused weights $b_i$ as the barycenter that minimizes the sum of UOT costs over observed modalities:

$$b_i \in \arg \min_{b \in \Delta_K} \sum_{v : m_i^v = 1} \lambda_i^v \, \mathrm{UOT}_{\varepsilon,\tau}(a_i^v, b; C_i^v). \tag{10}$$

Because $\beta_i$ always lives on the same prototype support, the fused representation has a fixed dimension and is comparable across samples. This also keeps each transport problem small: it is solved on a $J \times K$ grid instead of across long token sequences.

The weights $\lambda_i^v$ control modality importance among the observed set. We tie them to the transport solution so that a modality receives less weight when a large fraction of its mass is effectively trimmed. Concretely, once we have a transport plan $\pi_i^v$, we define the transported mass

$$t_i^v = \|\pi_i^v \mathbf{1}\|_1, \tag{11}$$

which is at most 1 under the KL-relaxed formulation. We also allow a small learned gate that depends on the modality content:

$$s_i^v = \sigma(h(\mathrm{Pool}(Q_i^v))), \tag{12}$$

where $\mathrm{Pool}(\cdot)$ is mean pooling over atoms, $h$ is a two-layer MLP shared across modalities, and $\sigma$ is the sigmoid. We then set

$$\lambda_i^v = \frac{m_i^v \, (t_i^v s_i^v)}{\sum_{u : m_i^u = 1} (t_i^u s_i^u) + \delta}, \tag{13}$$

with a small $\eta > 0$ for stability. This keeps $\lambda_i^v$ data-dependent, but it remains anchored to the trimming behavior of UOT rather than acting as an unconstrained attention module.

## 3.5. Differentiable solver and training

We solve (10) with generalized Sinkhorn iterations and unroll a small number of steps so the fusion module is end-to-end differentiable. For modality $v$ and sample $i$, define the Gibbs kernel

$$K_i^v = \exp(-C_i^v/\varepsilon). \tag{14}$$

We parameterize the plan as $\pi_i^v = \text{diag}(u_i^v)\, K_i^v \, \text{diag}(v_i^v)$ with scalings $u_i^v \in \mathbb{R}_+^J$ and $v_i^v \in \mathbb{R}_+^K$. Let

$$\theta = \frac{\tau}{\tau + \varepsilon}. \tag{15}$$

Note that $\theta$ is a *derived* quantity rather than an independent hyperparameter. Although $\theta \in [0,1]$ alone controls the exponent in the Sinkhorn updates, the entropic scale $\varepsilon$ also enters the Gibbs kernel $K_i^v = \exp(-C_i^v/\varepsilon)$, so $(\tau, \varepsilon)$ cannot be collapsed into a single $\theta$ without losing control over regularization strength. We therefore parameterize the solver by $(\tau, \varepsilon)$ and report sensitivity in Table 6. Starting from $b_i$ uniform on $\Delta_K$ and $u_i^v, v_i^v$ initialized to ones, each iteration updates

$$u_i^v \leftarrow \left(\frac{a_i^v}{K_i^v v_i^v + \eta}\right)^{\theta}, \qquad v_i^v \leftarrow \left(\frac{b_i}{(K_i^v)^{\top} u_i^v + \eta}\right)^{\theta}, \tag{16}$$

where division and power are elementwise. We then update $b_i$ by a fixed-point step and renormalize to the simplex:

$$b_i \leftarrow \text{Normalize}\!\left(\left(\sum_{v:m_i^v=1} \lambda_i^v \big((K_i^v)^{\top} u_i^v\big)^{\theta}\right)^{\frac{1}{\theta}}\right),$$
$$\text{Normalize}(z) = \frac{z}{\max(\|z\|_1, \, \eta)}. \tag{17}$$

The $\max$ with $\eta$ safeguards against underflow when all entries of $z$ are vanishingly small; whenever $\|z\|_1 \geq \eta$ (which holds throughout training in practice), this reduces to the exact simplex projection $z/\|z\|_1$ and $b_i \in \Delta_K$ strictly. We use a log-stabilized implementation of the matrix-vector products in (16) when needed, but the equations above capture the solver that is unrolled in training.

The fusion output $b_i$ is fed into a classifier $c(\cdot)$, and we train all components end-to-end with cross-entropy:

$$\mathcal{L} = \frac{1}{N} \sum_{i=1}^{N} \text{CE}\big(c(b_i), y_i\big). \tag{18}$$

Here $c(\cdot): \Delta_K \to \mathbb{R}^C$ is a lightweight classification head (a two-layer MLP) that maps the fused prototype measure $b_i$ to class logits over the $C$ action classes. Cross-entropy is computed against the ground-truth class label $y_i$, *not* against the prototype indices: the prototypes $\{p_k\}_{k=1}^{K}$ are never directly supervised and are learned end-to-end as an intermediate shared support such that the induced $b_i$

is discriminative for classification. Consequently, $K$ is a capacity hyperparameter of the latent factorization and need not correspond to the number of classes. When a modality is missing ($m_i^v = 0$), we skip its atom extraction and transport terms, so the module naturally handles arbitrary missing patterns at both training and test time.

Finally, the robustness of this design is limited by the learned geometry. If a corrupted modality is mapped close to the prototypes, transport has no reason to trim it. The barycenter can also fail when all observed modalities drift in the same wrong direction, since there is no clean signal to anchor prototype learning. These cases motivate the evaluation protocols in Section 5.

## 4. Theoretical Analysis

This section clarifies what the KL-relaxed UOT term guarantees, and what it does not. We do not aim to prove generalization; instead, we formalize a bounded-cost trimming property implied by Eq. (8). We adopt the convention $0 \log 0 = 0$ in the entropy term.

### 4.1. A bounded-cost trimming property

We consider the UOT objective used in Eq. (8). For convenience we restate it here. Given $a \in \Delta_J$, $b \in \Delta_K$, a cost matrix $C \in \mathbb{R}^{J \times K}$, and parameters $\varepsilon \geq 0$ and $\tau > 0$,

$$\text{UOT}_{\varepsilon,\tau}(a,b;C) = \min_{\pi \in \mathbb{R}_+^{J \times K}} \langle C, \pi \rangle + \varepsilon \sum_{j,k} \pi_{jk}(\log \pi_{jk} - 1)$$
$$+ \tau \, \text{KL}(\pi \mathbf{1} \,\|\, a) + \tau \, \text{KL}(\pi^{\top} \mathbf{1} \,\|\, b) \tag{19}$$

where KL is the generalized KL divergence in Eq. (9).

The next proposition states a simple but important fact: the objective can always avoid paying an arbitrarily large transport cost by leaving a part of the mass unmatched, and the price of doing so is controlled by $\tau$.

**Proposition 4.1** (Dropping mass has bounded penalty). *Fix $a \in \Delta_J$ and $b \in \Delta_K$. Let $S \subseteq \{1, \ldots, J\}$ be any subset of atoms, and let $\alpha = \sum_{j \notin S} a_j$ be the total mass outside $S$. For any cost matrix $C$, there exists a transport plan $\pi$ such that*

$$\text{UOT}_{\varepsilon,\tau}(a,b;C) \leq \sum_{j \in S} \sum_{k=1}^{K} a_j b_k \, C_{jk}$$
$$+ \varepsilon \sum_{j \in S} \sum_{k=1}^{K} a_j b_k \big(\log(a_j b_k) - 1\big)$$
$$+ 2\tau\alpha. \tag{20}$$

*In particular, the contribution of atoms in $\{1, \ldots, J\} \setminus S$ to the upper bound depends only on their total mass $\alpha$, and not on their distances in $C$.*

*Proof.* Define a candidate plan $\pi \in \mathbb{R}_+^{J \times K}$ by

$$\pi_{jk} = \begin{cases} a_j b_k, & j \in S, \\ 0, & j \notin S. \end{cases} \quad (21)$$

Its row sums are $(\pi \mathbf{1})_j = a_j$ for $j \in S$ and $(\pi \mathbf{1})_j = 0$ for $j \notin S$. Its column sums are $\pi^\top \mathbf{1} = (1 - \alpha)b$.

The transport term and the entropy term under this $\pi$ are exactly the first two terms on the right-hand side of Eq. (20). The remaining part comes from the KL penalties. For the row KL term, using the definition of generalized KL, each dropped row contributes

$$\mathrm{KL}(0 \,\|\, a_j) = a_j, \quad (22)$$

while rows in $S$ contribute zero because they match $a_j$ exactly. Hence

$$\mathrm{KL}(\pi \mathbf{1} \,\|\, a) = \sum_{j \notin S} a_j = \alpha. \quad (23)$$

For the column KL term, since $\pi^\top \mathbf{1} = (1 - \alpha)b$ and $\sum_k b_k = 1$,

$$\mathrm{KL}\big((1-\alpha)b \,\|\, b\big) = \sum_{k=1}^{K} \big((1-\alpha)b_k \log(1-\alpha) - (1-\alpha)b_k + b_k\big)$$
$$= \alpha + (1 - \alpha) \log(1 - \alpha) \le \alpha. \quad (24)$$

because $(1 - \alpha) \log(1 - \alpha) \le 0$ for $\alpha \in [0, 1]$. Multiplying by $\tau$ and summing the two KL terms gives at most $2\tau\alpha$. Since $\mathrm{UOT}_{\varepsilon,\tau}$ is the minimum over all $\pi \ge 0$, plugging this candidate plan yields Eq. (20). □

Proposition 4.1 explains why UOT can be stable under severe corruption. If a subset of atoms is pushed far away in the latent space, balanced matching would make the objective grow with the distance. Here the model can reduce the influence of those atoms by leaving them largely unmatched. The price is at most linear in their total mass, scaled by $\tau$. This is the mechanism we rely on when a modality contains partial failures.

# 5. Experiments

We evaluate our method on multi-modal action recognition under *missing* and *corrupted* views. The goal is to test whether unbalanced optimal transport (UOT) yields robust fusion when the observation process is unreliable. We first describe the datasets and evaluation protocol, then compare with existing methods, and finally analyze the trimming behavior directly.

## 5.1. Experimental Setup

**Datasets.** We use three action recognition benchmarks. **NTU RGB+D 60** (Shahroudy et al., 2016) contains 56,880 clips across 60 action classes with RGB, skeleton, and infrared modalities. **NTU RGB+D 120** (Liu et al., 2019) extends this to 114,480 clips and 120 classes.

We report Top-1 accuracy under standard Cross-Subject (X-Sub), Cross-View (X-View), and Cross-Setup (X-Set) splits. For NTU datasets, we use RGB, skeleton, and infrared as three modalities when available.

**Implementation details.** We implement the model in Py-Torch. We use pre-trained CTR-GCN (Chen et al., 2021) for skeleton and Video Swin-B (Liu et al., 2022) for RGB. For UOT fusion, we set $\varepsilon = 0.1$, $\tau = 0.5$, $J = K = 16$, and run $T = 20$ generalized Sinkhorn iterations. We train for 50 epochs using Adam with learning rate $10^{-4}$ and batch size 64. All experiments use one RTX 3090 GPU with FP32 precision.

**Missing protocol and evaluation setup.** Let $V$ be the number of modalities (or views). For a nominal missing rate $r \in [0, 1]$, we generate a binary mask $m_i \in \{0, 1\}^V$ for each sample $i$ by dropping each modality independently:

$$m_i^v \sim \mathrm{Bernoulli}(1 - r), \quad v = 1, \ldots, V.$$

To avoid undefined fusion when all modalities are missing, we apply the following rule: if $\sum_{v=1}^{V} m_i^v = 0$, we uniformly select one modality $v^\star$ and set $m_i^{v^\star} = 1$. Thus every sample has at least one observed modality.

This rule changes the *effective* missingness slightly. In particular, the expected number of observed modalities becomes

$$\mathbb{E}\Big[ \sum_{v=1}^{V} m_i^v \Big] = V(1 - r) + r^V.$$

For example, with $V = 3$ and $r = 0.8$, $\mathbb{E}[\#\text{observed}] = 3 \times 0.2 + 0.8^3 = 1.112$. Equivalently, the per-modality probability of being observed becomes $(1 - r) + r^V/V$, which is 0.3707 in this setting (i.e., an effective per-modality missing rate of about 62.9%). Following common practice, we still report results under the nominal rate $r$ for ease of comparison, and we use the *same* missing-mask generation procedure for all multi-modal baselines for fairness. Unless stated otherwise, we apply the same protocol at test time.

**Baselines.** We compare against three groups of baselines under the same missing-mask generation protocol.

**(1) Multi-modal action recognition.** We include *Late Fusion*, *Channel Exchange* (Wang et al., 2020b), and robust multi-modal models *MLA* (Zhang et al., 2024) and *DMR-Net* (Wei et al., 2024). We also compare recent 2025 meth-

ods: *RCMCL* (Akgul et al., 2025) (self-supervised pretraining then supervised fine-tuning), *CalMRL* (Liu et al., 2025) (plug-and-play calibration module, [†]), and *MV-GMN* (Lin et al., 2025) (aggregation module with our backbones).

**(2) Missing-view learning (adapted).** We adapt *DCP* (Lin et al., 2023), *CPSPAN* (Jin et al., 2023), and *UIMC* (Xie et al., 2023) by replacing their prediction heads with temporal pooling and a linear classifier. Methods implemented by us are marked [†].

**(3) Single-modality references.** We report unimodal backbones *CTR-GCN* (Chen et al., 2021) (skeleton), *Video Swin-B* (Liu et al., 2022) (RGB), and *BHaRNet* (Cho & Kim, 2025) (skeleton). These assume full modality availability and serve as upper bounds rather than missing-view baselines.

## 5.2. Performance Under Missing Views

Table 1 reports accuracy under varying missing rates on NTU 60 and NTU 120. At low missing rates (0–20%), recent robust methods such as RCMCL and MLA are competitive; in fact, at $r{=}0$ RCMCL marginally outperforms our method on both splits (e.g., 96.2 vs. 96.0 on NTU 60 X-Sub), which is consistent with our positioning that UOT-Fusion targets robustness rather than universal gains under clean observations. As the missing rate increases, the performance gap widens in our favor: under 80% nominal missing on NTU 60 X-Sub, our method reaches 78.3% while the strongest 2025 baseline RCMCL achieves 70.8% (a +7.5-point gap), and MLA/DMRNet degrade to 68.9%/68.2%. We attribute this widening advantage to the trimming behavior of UOT: when a modality is unreliable (or only partially informative), the UOT objective can reduce its influence by leaving a fraction of its mass unmatched, whereas balanced aggregation must always explain every observed input, which amplifies errors under heavy missingness. Single-modality references provide additional context: CTR-GCN alone reaches 92.4% on NTU 60 when skeleton is always available, and Video Swin-B reaches 94.2% when RGB is always available. Under the $V = 3$, $r = 0.8$ protocol, the expected number of observed modalities is about 1.11 per sample, so the fusion accuracy around 78% is plausible given that the available modality varies across samples and fusion must operate under heterogeneous missingness.

We visualize the performance trends and the corresponding transported mass in Figure 1, demonstrating how our UOT-Fusion adaptively reduces the influence of unreliable views as corruption intensity or missing rates increase.

## 5.3. Generalization Under Missing-Rate Shift

We further test robustness when the test-time missing rate differs from training. In Table 2, we train all models at

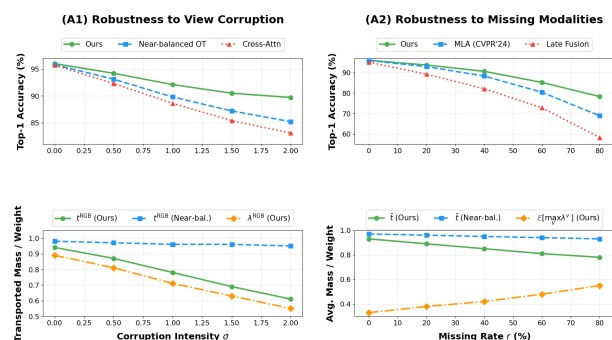

*Figure 1.* (A1-A2) Top-1 accuracy and transported mass under varying corruption and missing rates.

a fixed 20% missing rate and evaluate across a range of test-time missing rates. As test-time missingness increases, all methods degrade, but the degradation is smaller for our approach. For example, from 0% to 80% test missing, Late Fusion drops from 91.3% to 60.1% (31.2 percentage points), MLA drops from 94.0% to 70.5% (23.5 points), while our method drops from 94.9% to 79.8% (15.1 points). This is consistent with the intended behavior of UOT: the fusion relies on matching costs and can reduce the influence of less compatible inputs, which remains applicable under shifted missingness.

## 5.4. View Corruption and Adaptive Trimming

Missingness is not the only failure mode; a modality can be present but corrupted. To test this directly, we conduct a controlled experiment where all modalities are present, but one modality is corrupted with additive Gaussian noise. Specifically, for each test sample, we add noise $\mathcal{N}(0, \sigma^2 I)$ to the *encoded latent features* of the RGB modality (after the encoder and before atom extraction). We vary $\sigma \in \{0, 0.5, 1.0, 1.5, 2.0\}$ and measure accuracy and the transported mass $t^{\mathrm{RGB}}$ for RGB, as defined in Section 3.4.

Table 3 shows results on NTU 60 X-Sub. As corruption increases, our method degrades more gracefully (from 96.0% to 89.7%), while the transported mass $t^{\mathrm{RGB}}$ decreases (from 0.94 to 0.61), indicating that the solver downweights the corrupted modality by leaving part of its mass unmatched. We also include two reference fusions: (i) a cross-attention fusion baseline (a lightweight Transformer-style fusion block) and (ii) a near-balanced OT baseline implemented by using a large marginal penalty (we set $\tau = 10$), which yields transported mass close to 1.0. These baselines degrade faster under corruption, consistent with the lack of explicit trimming.

Finally, this experiment highlights a limitation: trimming depends on geometry in the learned embedding space. If

*Table 1.* Top-1 accuracy (%) under varying missing rates. Results are mean $\pm$ std over 3 runs with different random seeds. Best in **bold**, second underlined. Methods marked with [†] are implemented by us.

| 2*Method | NTU 60 (X-Sub) Missing Rate | | | | | NTU 120 (X-Set) Missing Rate | | | | |
|---|---|---|---|---|---|---|---|---|---|---|
| | 0% | 20% | 40% | 60% | 80% | 0% | 20% | 40% | 60% | 80% |
| *Single-Modality References (modality always available)* | | | | | | | | | | |
| CTR-GCN (Skel) (Chen et al., 2021) | 92.4 | 92.4 | 92.4 | 92.4 | 92.4 | 88.9 | 88.9 | 88.9 | 88.9 | 88.9 |
| BHaRNet (Skel) (Cho & Kim, 2025) | 93.2 | 93.2 | 93.2 | 93.2 | 93.2 | 89.6 | 89.6 | 89.6 | 89.6 | 89.6 |
| Video Swin-B (RGB) (Liu et al., 2022) | 94.2 | 94.2 | 94.2 | 94.2 | 94.2 | 91.5 | 91.5 | 91.5 | 91.5 | 91.5 |
| *Multi-Modal Fusion (all modalities subject to missing)* | | | | | | | | | | |
| Late Fusion | $95.1_{\pm0.3}$ | $89.2_{\pm0.5}$ | $82.1_{\pm0.7}$ | $72.8_{\pm0.9}$ | $58.3_{\pm1.2}$ | $92.7_{\pm0.4}$ | $86.3_{\pm0.6}$ | $77.9_{\pm0.8}$ | $67.2_{\pm1.0}$ | $52.4_{\pm1.3}$ |
| Channel Exchange (Wang et al., 2020b) | $95.2_{\pm0.2}$ | $90.1_{\pm0.4}$ | $83.5_{\pm0.6}$ | $74.8_{\pm0.8}$ | $61.2_{\pm1.1}$ | $93.0_{\pm0.3}$ | $87.1_{\pm0.5}$ | $79.3_{\pm0.7}$ | $69.1_{\pm0.9}$ | $54.8_{\pm1.2}$ |
| *Incomplete / Missing-view Learning (adapted to action recognition)* | | | | | | | | | | |
| DCP (Lin et al., 2023) | $95.7_{\pm0.3}$ | $92.3_{\pm0.4}$ | $87.2_{\pm0.5}$ | $79.6_{\pm0.7}$ | $68.9_{\pm0.9}$ | $93.4_{\pm0.3}$ | $89.8_{\pm0.5}$ | $83.7_{\pm0.6}$ | $75.2_{\pm0.8}$ | $62.9_{\pm1.0}$ |
| CPSPAN[†] (Jin et al., 2023) | $95.3_{\pm0.4}$ | $91.8_{\pm0.5}$ | $86.5_{\pm0.6}$ | $78.7_{\pm0.8}$ | $67.4_{\pm1.0}$ | $92.9_{\pm0.4}$ | $89.2_{\pm0.5}$ | $82.9_{\pm0.7}$ | $74.3_{\pm0.9}$ | $61.5_{\pm1.1}$ |
| UIMC (Xie et al., 2023) | $95.6_{\pm0.3}$ | $92.1_{\pm0.4}$ | $86.9_{\pm0.5}$ | $79.2_{\pm0.7}$ | $68.5_{\pm0.9}$ | $93.2_{\pm0.3}$ | $89.6_{\pm0.4}$ | $83.4_{\pm0.6}$ | $74.9_{\pm0.8}$ | $62.6_{\pm1.0}$ |
| CalMRL[†] (Liu et al., 2025) | $95.8_{\pm0.3}$ | $92.6_{\pm0.4}$ | $87.8_{\pm0.5}$ | $80.7_{\pm0.7}$ | $70.1_{\pm0.9}$ | $93.6_{\pm0.3}$ | $90.2_{\pm0.4}$ | $84.6_{\pm0.6}$ | $76.8_{\pm0.8}$ | $65.2_{\pm1.0}$ |
| *Recent Robust Multi-Modal Methods (2025)* | | | | | | | | | | |
| RCMCL (Akgul et al., 2025) | **96.2**$_{\pm0.2}$ | 93.1$_{\pm0.3}$ | 88.6$_{\pm0.4}$ | 81.2$_{\pm0.6}$ | 70.8$_{\pm0.8}$ | **94.3**$_{\pm0.3}$ | 91.1$_{\pm0.4}$ | 85.8$_{\pm0.5}$ | 77.4$_{\pm0.7}$ | 66.3$_{\pm0.9}$ |
| MV-GMN (Lin et al., 2025) | $95.8_{\pm0.3}$ | $92.7_{\pm0.4}$ | $88.1_{\pm0.5}$ | $80.6_{\pm0.6}$ | $69.5_{\pm0.8}$ | $93.8_{\pm0.3}$ | $90.5_{\pm0.4}$ | $85.1_{\pm0.5}$ | $76.7_{\pm0.7}$ | $64.9_{\pm0.9}$ |
| *Recent Robust Multi-Modal Methods (2024)* | | | | | | | | | | |
| MLA (CVPR'24) (Zhang et al., 2024) | $95.9_{\pm0.2}$ | $92.9_{\pm0.3}$ | $88.3_{\pm0.5}$ | $80.4_{\pm0.6}$ | $68.9_{\pm0.8}$ | $93.7_{\pm0.3}$ | $90.4_{\pm0.4}$ | $84.9_{\pm0.5}$ | $76.3_{\pm0.7}$ | $63.8_{\pm0.9}$ |
| DMRNet (ECCV'24) (Wei et al., 2024) | $95.7_{\pm0.3}$ | $92.5_{\pm0.4}$ | $87.9_{\pm0.5}$ | $79.8_{\pm0.6}$ | $68.2_{\pm0.8}$ | $93.5_{\pm0.3}$ | $90.1_{\pm0.4}$ | $84.3_{\pm0.5}$ | $75.8_{\pm0.7}$ | $64.1_{\pm0.9}$ |
| **Ours (UOT-Fusion)** | 96.0$_{\pm0.2}$ | **93.7**$_{\pm0.3}$ | **90.6**$_{\pm0.4}$ | **85.2**$_{\pm0.5}$ | **78.3**$_{\pm0.7}$ | 94.1$_{\pm0.2}$ | **91.8**$_{\pm0.3}$ | **87.9**$_{\pm0.4}$ | **82.1**$_{\pm0.6}$ | **74.6**$_{\pm0.8}$ |
| $\Delta$ vs. RCMCL | $-0.2$ | $+0.6$ | $+2.0$ | $+4.0$ | $+7.5$ | $-0.2$ | $+0.7$ | $+2.1$ | $+4.7$ | $+8.3$ |

the encoder maps corrupted inputs close to clean prototypes, transport costs remain low and trimming may not activate.

Figure 2 provides a sample-level visualization of the transport plans, illustrating how the model maintains robustness by producing sparse, trimmed plans for corrupted inputs in successful cases.

### 5.5. Ablation Study

Table 4 validates each design component at $40\%$ missing rate on NTU 60 X-Sub. Replacing UOT with cross-attention fusion causes a 7.6 percentage-point drop, indicating that forcing all available content to participate in fusion can be harmful under unreliable observations. Using near-balanced OT ($\tau = 10$) improves over attention but still lags by 3.6 points, consistent with the inability to discard outlier mass. Removing the prototype support and fusing directly in feature space drops performance by 3.2 points, suggesting that the shared prototype support stabilizes matching. Finally, using fixed uniform weights $\lambda$ instead of the mass-dependent weighting in Eq. (13) decreases accuracy by 1.3 points. The full model combines all components and achieves $90.6\%$.

### 5.6. Computational Cost

Table 5 reports inference time and parameter count. We measure end-to-end inference on a single sample (batch size 1) on an RTX 3090 with FP32 precision, averaged over 1000 samples. Timing includes the backbone forward pass and the fusion module (including $T = 20$ Sinkhorn iterations for OT-based methods, when applicable).

The backbone encoders (CTR-GCN and Video Swin-B) take 16.4ms and have 82.3M parameters. Adding our UOT fusion with $J = K = 16$ increases time to 18.2ms (1.8ms overhead, $\sim 11\%$ relative increase) and parameters to 83.1M (0.8M added for prototypes and lightweight fusion/classification layers). Increasing to $J = K = 32$ yields a marginal accuracy gain (+0.3 points) but increases fusion time.

**Asymptotic complexity.** The fusion module's theoretical cost per sample is $\mathcal{O}(T \cdot V \cdot JK)$ for the Sinkhorn-style unrolled solver, where $T$ is the number of iterations, $V$ the number of observed modalities, and $J, K$ the atom and prototype counts. With $T=20$, $V \leq 3$, $J=K=16$, each fusion forward pass requires on the order of $1.5 \times 10^4$ kernel operations, negligible compared to the $\sim 10^{10}$ FLOPs of the backbones (CTR-GCN + Video Swin-B). For comparison, balanced entropic OT fusion shares the same asymptotic cost but lacks mass trimming; cross-attention fusion across raw tokens scales as $\mathcal{O}(V \cdot n_v \cdot J)$ in the token length $n_v$, which is typically far larger than $K$ (e.g., $n_v \approx 196$ for Swin-B vs. $K=16$). This explains why UOT-Fusion introduces only $\sim 11\%$ wall-clock overhead in Table 5 while delivering substantially better robustness.

### 5.7. Hyperparameter Sensitivity

Table 6 analyzes sensitivity to UOT parameters $\tau$ and $\varepsilon$ on NTU 60 at $40\%$ missing. Performance is stable within $\tau \in [0.2, 1.0]$ and $\varepsilon \in [0.05, 0.2]$. When $\tau \leq 0.1$, marginal relaxation becomes too loose and the model may over-trim,

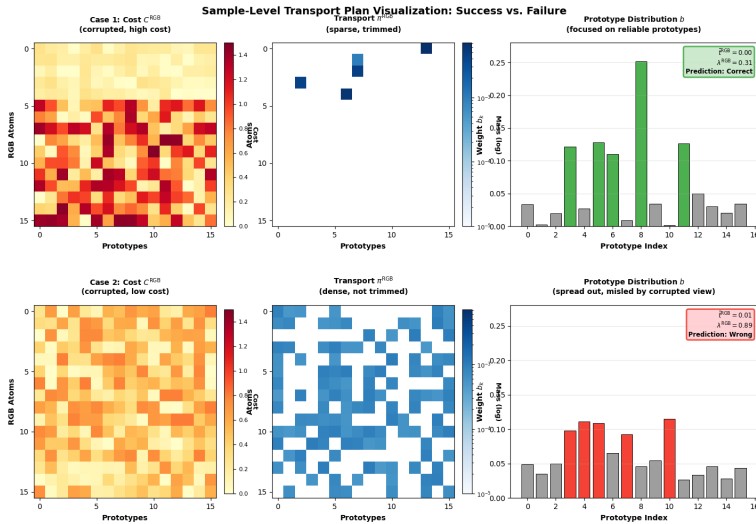

*Figure 2.* Visualization of cost matrices and transport plans for success and failure cases under corruption.

*Table 2.* Generalization under missing-rate shift on NTU 60 (X-Sub). All multi-modal models are trained with 20% missing and tested at various missing rates. Numbers are mean ± std over 3 runs.

| Method | Test Missing Rate | | | | | Drop (0→80) |
|---|---|---|---|---|---|---|
| | 0% | 20% | 40% | 60% | 80% | |
| Late Fusion | $91.3_{\pm0.4}$ | $89.2_{\pm0.5}$ | $82.7_{\pm0.7}$ | $74.2_{\pm0.9}$ | $60.1_{\pm1.2}$ | 31.2 |
| Channel Exchange (Wang et al., 2020b) | $92.1_{\pm0.3}$ | $90.1_{\pm0.4}$ | $84.1_{\pm0.6}$ | $75.8_{\pm0.8}$ | $61.7_{\pm1.1}$ | 30.4 |
| DCP (Lin et al., 2023) | $93.4_{\pm0.3}$ | $92.3_{\pm0.4}$ | $87.9_{\pm0.5}$ | $79.9_{\pm0.7}$ | $70.3_{\pm0.9}$ | 23.1 |
| CPSPAN[†] (Jin et al., 2023) | $93.0_{\pm0.4}$ | $91.8_{\pm0.5}$ | $87.2_{\pm0.6}$ | $79.1_{\pm0.8}$ | $69.8_{\pm1.0}$ | 23.2 |
| CalMRL[†] (Liu et al., 2025) | $93.6_{\pm0.3}$ | $92.6_{\pm0.4}$ | $88.4_{\pm0.5}$ | $81.5_{\pm0.7}$ | $71.3_{\pm0.9}$ | 22.3 |
| RCMCL (Akgul et al., 2025) | $\underline{94.3}_{\pm0.3}$ | $\underline{93.1}_{\pm0.3}$ | $\underline{89.1}_{\pm0.4}$ | $\underline{82.1}_{\pm0.6}$ | $\underline{71.6}_{\pm0.8}$ | 22.7 |
| MV-GMN (Lin et al., 2025) | $93.8_{\pm0.3}$ | $92.7_{\pm0.4}$ | $88.6_{\pm0.5}$ | $81.1_{\pm0.6}$ | $70.8_{\pm0.8}$ | 23.0 |
| MLA (CVPR'24) (Zhang et al., 2024) | $94.0_{\pm0.3}$ | $92.9_{\pm0.3}$ | $88.7_{\pm0.5}$ | $81.3_{\pm0.6}$ | $70.5_{\pm0.8}$ | 23.5 |
| DMRNet (ECCV'24) (Wei et al., 2024) | $93.7_{\pm0.3}$ | $92.5_{\pm0.4}$ | $88.3_{\pm0.5}$ | $80.6_{\pm0.7}$ | $69.8_{\pm0.9}$ | 23.9 |
| **Ours (UOT-Fusion)** | $\mathbf{94.9}_{\pm0.2}$ | $\mathbf{93.7}_{\pm0.3}$ | $\mathbf{90.9}_{\pm0.4}$ | $\mathbf{86.4}_{\pm0.5}$ | $\mathbf{79.8}_{\pm0.7}$ | 15.1 |

discarding useful signal. When $\tau > 2.0$, the objective approaches near-balanced matching and robustness decreases. Extreme $\varepsilon$ values can also lead to numerical instability in the Sinkhorn solver. We use $\tau = 0.5$ and $\varepsilon = 0.1$ throughout.

### 5.8. Limitations and Failure Modes

Our method adopts a deliberately geometric notion of reliability, trimming evidence only when it becomes expensive to match in the learned embedding space. While this design provides robustness to distant contamination, it also implies clear failure modes. If a corrupted modality is mapped close to the prototype support—or to an incorrect region with low transport cost—the trimming mechanism can become effectively inactive, and the fused barycenter may be confidently wrong. More generally, the approach assumes that at least part of the observed evidence remains anchored to a correct region of the prototype space; when

all modalities drift consistently due to systematic bias, synchronized failure, or shared spurious cues, the barycenter can become stable yet wrong, and unbalanced transport cannot recover a correct reference. Anchoring fusion on a fixed prototype support further introduces a strong inductive bias: rare but correct patterns that are far from the learned prototypes (e.g., long-tail motions or unusual viewpoints) may be treated as geometrically incompatible and trimmed, even though they carry useful signal. Finally, although features are normalized and transport hyperparameters are analyzed empirically, the practical behavior remains sensitive to cost scaling, marginal relaxation, and solver accuracy, with overly large or small trimming strength leading to loss of robustness or over-pruning. Overall, these limitations reflect a fundamental property of the objective: transport can only judge reliability through geometry, and failures that are not separable in the embedding space remain inherently

*Table 3.* Robustness to view corruption on NTU 60 (X-Sub). All modalities are observed, but RGB is corrupted with Gaussian noise $\mathcal{N}(0, \sigma^2 I)$ added to latent features. We report accuracy and transported mass $t^{\mathrm{RGB}}$ for the corrupted modality.

| 2*Method | Noise level $\sigma$ | | | | |
|---|---|---|---|---|---|
| | 0.0 | 0.5 | 1.0 | 1.5 | 2.0 |
| *Accuracy (%)* | | | | | |
| Cross-Attn Fusion | 95.7 | 92.3 | 88.6 | 85.4 | 83.1 |
| Near-Balanced OT ($\tau = 10$) | 95.8 | 93.1 | 89.8 | 87.2 | 85.2 |
| **Ours (UOT-Fusion)** | **96.0** | **94.2** | **92.1** | **90.5** | **89.7** |
| *Transported mass $t^{\mathrm{RGB}}$ for corrupted RGB modality* | | | | | |
| Near-Balanced OT ($\tau = 10$) | 0.98 | 0.97 | 0.96 | 0.96 | 0.95 |
| **Ours (UOT-Fusion)** | **0.94** | **0.87** | **0.78** | **0.69** | **0.61** |

*Table 4.* Ablation study on NTU 60 (X-Sub) at $40\%$ missing rate. Each row removes or modifies one component.

| Configuration | Accuracy (%) |
|---|---|
| **Full model (UOT + Prototypes + Adaptive $\lambda$)** | $\mathbf{90.6}_{\pm 0.4}$ |
| (a) w/o UOT (cross-attention fusion) | $83.0_{\pm 0.6}$ |
| (b) Near-balanced OT ($\tau = 10$) | $87.0_{\pm 0.5}$ |
| (c) w/o Prototypes (direct feature fusion) | $87.4_{\pm 0.5}$ |
| (d) Fixed uniform $\lambda$ | $89.3_{\pm 0.4}$ |
| Oracle (no missing, all modalities observed) | $96.0_{\pm 0.2}$ |

*Table 5.* Computational cost on NTU 60 (40% missing). Inference time is measured on RTX 3090, FP32, batch size 1, averaged over 1000 samples. Time includes backbone forward pass and the fusion module.

| Method | Params (M) | GFLOPs | Time (ms) | Acc. (%) |
|---|---|---|---|---|
| Backbones only | 82.3 | 45.2 | 16.4 | – |
| + Late Fusion | 82.3 | 45.2 | 16.5 | 82.1 |
| + Channel Exchange (Wang et al., 2020b) | 86.7 | 48.1 | 17.9 | 83.5 |
| + DCP (Lin et al., 2023) | 84.5 | 46.8 | 18.4 | 87.2 |
| + MLA (Zhang et al., 2024) | 87.0 | 49.3 | 19.2 | 88.3 |
| **+ Ours** ($J=K=16$) | 83.1 | 46.3 | 18.2 | **90.6** |
| **+ Ours** ($J=K=32$) | 83.4 | 47.1 | 19.6 | 90.9 |

*Table 6.* Sensitivity to UOT hyperparameters on NTU 60 (X-Sub, 40% missing). Default: $\tau=0.5$, $\varepsilon=0.1$.

| $\tau$ | Acc. (%) | $\varepsilon$ | Acc. (%) |
|---|---|---|---|
| 0.05 | 85.7 | 0.01 | 88.3 |
| 0.1 | 88.6 | 0.05 | 89.9 |
| 0.2 | 89.8 | 0.1 | **90.6** |
| 0.5 | **90.6** | 0.2 | 90.3 |
| 1.0 | 90.2 | 0.5 | 89.1 |
| 2.0 | 88.1 | 1.0 | 87.4 |

## Acknowledgements

We thank the anonymous reviewers for their constructive feedback, which helped improve this paper.

## Impact Statement

This paper advances the field of machine learning by improving the robustness of multi-view fusion under missing and corrupted observations. Potential downstream applications include assistive healthcare, human–robot interaction, and industrial safety monitoring. At the same time, action-recognition technology built on RGB video carries well-known risks related to privacy and surveillance; we encourage practitioners to favor privacy-preserving modalities (e.g., skeleton or infrared), adopt on-device inference, and comply with applicable data protection regulations. We also caution that our trimming mechanism judges reliability through geometry in a learned embedding space, so corrupted inputs mapped close to the prototype support may be silently accepted; the module should therefore not be used as a standalone safety detector. Our experiments rely solely on publicly available benchmarks and do not involve new human-subject data collection.

## References

Akgul, H., Eplik, M., Rojas, J., Yamamoto, A., Kumar, R., and Singh, M. Rcmcl: A unified contrastive learning framework for robust multi-modal (rgb-d, skeleton, point cloud) action understanding. *arXiv preprint*

challenging.

## 6. Conclusion

We presented a robust multi-view fusion module based on prototype-anchored unbalanced optimal transport. By relaxing marginal constraints with a generalized KL penalty, our approach allows the model to adaptively trim unreliable or missing evidence when matching costs are high. This mechanism provides a principled, differentiable alternative to standard aggregation methods that often struggle with corrupted inputs. Theoretical analysis confirms that the influence of outliers is bounded by their mass, independent of their distance in the latent space. Extensive experiments on action recognition benchmarks demonstrate that our method maintains high stability under severe missingness and feature-space corruption with minimal computational overhead. This framework offers a scalable and robust solution for multimodal learning in real-world environments where observations are frequently imperfect.

This paper presents work whose goal is to advance the field of machine learning. There are many potential societal consequences of our work, none of which we feel must be specifically highlighted here.

*arXiv:2511.04351*, 2025.

Baltrušaitis, T., Ahuja, C., and Morency, L.-P. Multimodal machine learning: A survey and taxonomy. *IEEE transactions on pattern analysis and machine intelligence*, 41 (2):423–443, 2018.

Chen, Y., Zhang, Z., Yuan, C., Li, B., Deng, Y., and Hu, W. Channel-wise topology refinement graph convolution for skeleton-based action recognition. In *Proceedings of the IEEE/CVF international conference on computer vision*, pp. 13359–13368, 2021.

Chizat, L., Peyré, G., Schmitzer, B., and Vialard, F.-X. An interpolating distance between optimal transport and fisher–rao metrics. *Foundations of Computational Mathematics*, 18(1):1–44, 2018.

Cho, S. and Kim, T.-K. Body-hand modality expertized networks with cross-attention for fine-grained skeleton action recognition. *arXiv preprint arXiv:2503.14960*, 2025.

Claici, S., Chien, E., and Solomon, J. Stochastic wasserstein barycenters. In *International Conference on Machine Learning*, pp. 999–1008. PMLR, 2018.

Cuturi, M. Sinkhorn distances: Lightspeed computation of optimal transport. *Advances in neural information processing systems*, 26, 2013.

Cuturi, M. and Doucet, A. Fast computation of wasserstein barycenters. In *International conference on machine learning*, pp. 685–693. PMLR, 2014.

Hendrycks, D. and Dietterich, T. Benchmarking neural network robustness to common corruptions and perturbations. *arXiv preprint arXiv:1903.12261*, 2019.

Hudson, D. A. and Manning, C. D. Compositional attention networks for machine reasoning. *arXiv preprint arXiv:1803.03067*, 2018.

Jaegle, A., Gimeno, F., Brock, A., Vinyals, O., Zisserman, A., and Carreira, J. Perceiver: General perception with iterative attention. In *International conference on machine learning*, pp. 4651–4664. PMLR, 2021.

Jiang, B., Xiang, J., Wu, X., Wang, Y., Chen, H., Cao, W., and Sheng, W. Robust multi-view learning via adaptive regression. *Information Sciences*, 610:916–937, 2022.

Jin, J., Wang, S., Dong, Z., Liu, X., and Zhu, E. Deep incomplete multi-view clustering with cross-view partial sample and prototype alignment. In *Proceedings of the IEEE/CVF conference on computer vision and pattern recognition*, pp. 11600–11609, 2023.

Liero, M., Mielke, A., and Savaré, G. Optimal entropy-transport problems and a new hellinger–kantorovich distance between positive measures. *Inventiones mathematicae*, 211(3):969–1117, 2018.

Lin, Y., Gou, Y., Liu, Z., Li, B., Lv, J., and Peng, X. Dual contrastive prediction for incomplete multi-view representation learning. volume 45, pp. 4447–4461. IEEE, 2023.

Lin, Y., Lu, J., Yong, Y., and Zhang, J. Mv-gmn: State space model for multi-view action recognition. *arXiv preprint arXiv:2501.13829*, 2025.

Liu, J., Shahroudy, A., Perez, M., Wang, G., Duan, L.-Y., and Kot, A. C. Ntu rgb+ d 120: A large-scale benchmark for 3d human activity understanding. *IEEE transactions on pattern analysis and machine intelligence*, 42(10): 2684–2701, 2019.

Liu, X., Xia, X., Wei, J., Yang, S., Su, X., Ng, S.-K., and Chua, T.-S. Calibrated multimodal representation learning with missing modalities. *arXiv preprint arXiv:2511.12034*, 2025.

Liu, Z., Ning, J., Cao, Y., Wei, Y., Zhang, Z., Lin, S., and Hu, H. Video swin transformer. In *Proceedings of the IEEE/CVF conference on computer vision and pattern recognition*, pp. 3202–3211, 2022.

Poklukar, P., Vasco, M., Yin, H., Melo, F. S., Paiva, A., and Kragic, D. Geometric multimodal contrastive representation learning. In *International Conference on Machine Learning*, pp. 17782–17800. PMLR, 2022.

Séjourné, T., Feydy, J., Vialard, F.-X., Trouvé, A., and Peyré, G. Sinkhorn divergences for unbalanced optimal transport. *arXiv preprint arXiv:1910.12958*, 2019.

Shahroudy, A., Liu, J., Ng, T.-T., and Wang, G. NTU RGB+D: A large scale dataset for 3D human activity analysis. In *Proceedings of the IEEE Conference on Computer Vision and Pattern Recognition*, pp. 1010–1019, 2016.

Solomon, J., De Goes, F., Peyré, G., Cuturi, M., Butscher, A., Nguyen, A., Du, T., and Guibas, L. Convolutional wasserstein distances: Efficient optimal transportation on geometric domains. *ACM Transactions on Graphics (ToG)*, 34(4):1–11, 2015.

Valada, A., Mohan, R., and Burgard, W. Self-supervised model adaptation for multimodal semantic segmentation. *International Journal of Computer Vision*, 128(5):1239–1285, 2020.

Vaswani, A., Shazeer, N., Parmar, N., Uszkoreit, J., Jones, L., Gomez, A. N., Kaiser, Ł., and Polosukhin, I. Attention is all you need. *Advances in neural information processing systems*, 30, 2017.

Wang, Q., Zhan, L., Thompson, P., and Zhou, J. Multimodal learning with incomplete modalities by knowledge distillation. In *Proceedings of the 26th ACM SIGKDD International Conference on Knowledge Discovery & Data Mining*, pp. 1828–1838, 2020a.

Wang, Y., Huang, W., Sun, F., Xu, T., Rong, Y., and Huang, J. Deep multimodal fusion by channel exchanging. *Advances in neural information processing systems*, 33: 4835–4845, 2020b.

Wei, S., Luo, Y., Wang, Y., and Luo, C. Robust multimodal learning via representation decoupling. In *European Conference on Computer Vision*, pp. 38–54. Springer, 2024.

Xie, M., Han, Z., Zhang, C., Bai, Y., and Hu, Q. Exploring and exploiting uncertainty for incomplete multi-view classification. In *Proceedings of the IEEE/CVF conference on computer vision and pattern recognition*, pp. 19873–19882, 2023.

Xu, C., Tao, D., and Xu, C. A survey on multi-view learning. *arXiv preprint arXiv:1304.5634*, 2013.

Zhang, X., Yoon, J., Bansal, M., and Yao, H. Multimodal representation learning by alternating unimodal adaptation. In *Proceedings of the IEEE/CVF conference on computer vision and pattern recognition*, pp. 27456–27466, 2024.

