# OpenReview forum: "Robust Multi-View Fusion via Prototype-Anchored Unbalanced Optimal Transport"
_ICML.cc/2026/Conference — ICML 2026 regular_

### Official Review · Reviewer_928L · 2026-03-03

**Soundness:** 3
**Presentation:** 3
**Significance:** 2
**Originality:** 2
**Overall Recommendation:** 5
**Confidence:** 4

**Summary:**

This paper studies robust multi-view or multi-modal fusion when some views are missing or corrupted. The research addresses a relevant challenge: standard fusion rules (e.g., attention/pooling or balanced alignment) tend to force every observed view to contribute, which can make the fused representation brittle under unreliable observations. The proposed solution is a prototype-anchored fusion module that computes an entropically-regularized unbalanced optimal transport (UOT) barycenter. Specifically, each view is compressed into a small set of learned “atoms” (via learned-query pooling), these atoms are matched to a shared set of learnable prototypes, and the fused output is a fixed-dimensional probability distribution over the prototypes. An important concept explored in the manuscript is that relaxing marginal constraints via generalized KL penalties enables differentiable “trimming”: if matching some atom mass to prototypes is geometrically expensive, the UOT objective can leave part of that mass unmatched with a bounded penalty, thereby downweighting unreliable evidence. The paper provides a simple theoretical bound, and experiments on multi-view action recognition benchmarks demonstrate improved stability under high missing rates, missing-rate shift, and feature corruption, with modest computational overhead.

**Compliance With Llm Reviewing Policy:**

Affirmed.

**Final Justification:**

Thank you for the detailed and very helpful rebuttal. I have read it carefully, and my concerns are now fully resolved. However, the paper still requires some revisions to be published. Therefore, I am increasing my overall recommendation from 4 (Weak Accept) to 5 (Accept).

**Key Questions For Authors:**

As described in the “Strength And Weakness” section.

**Limitations:**

As discussed in the “Strength and Weakness” section, the manuscript leverages unbalanced optimal transport to enhance robust multi-view fusion, but it still has several limitations, including unclear explanations, insufficient investigation of view corruption, lack of systematic analysis of cost matrix construction and prototype behavior. Moreover, the absence of experimental results on the NW-UCLA benchmark is particularly concerning, given its relevance as a multi-camera-view dataset (as opposed to multi-modality).

**Strengths And Weaknesses:**

Strength:
1.The proposed UOT-Fusion formulates multimodal fusion as an unbalanced optimal transport problem, achieving robust alignment and strong downstream performance while remaining simple and interpretable under view corruption scenarios.
2.The paper provides a theoretical analysis that explains the underlying mechanism of UOT-Fusion and clarifies why it works in practice.
Weakness:
1.Unclear training details: The paper does not clearly specify how the classification labels used for training the fusion module are determined. It is puzzling how the learnable prototype-based fusion—which is adapts during network training—can be supervised via cross-entropy with a probability vector over prototypes, raising questions about how ground-truth labels for such prototype assignments are defined.
2.Overly simplified view corruption in ablation studies: The ablation experiments only evaluate model performance under different levels of Gaussian noise applied to the RGB modality. The study lacks comparisons under a unified setting with other methods (e.g., recent robust multi-modal methods from 2025, single-modality baselines), and does not consider other forms of corruption such as occlusion, tracking failures, or corruption in other modalities.
3.Insufficient justification for cost function/similarity design: The behavior of UOT is almost entirely governed by the cost matrix. If the cost matrix is poorly chosen, it may either treat semantic differences as noise to be trimmed or absorb noise as low-cost matches. The paper does not sufficiently discuss the construction choices, scale normalization, or the impact of different cost designs on the trimming behavior and fusion outcomes.
4.Missing experiments on NW-UCLA benchmark: Although the NW-UCLA benchmark is mentioned in the “Dataset” section, no quantitative or qualitative results are reported for it—a notable omission given its multi-camera-view characteristic (as opposed to multi-modality).
5.Lack of analysis on prototype behavior: UOT-Fusion relies on a predefined set of prototypes, essentially assuming that different modalities share a common set of latent factors. While the paper explains how unbalanced transport contributes to robustness against corruption, it does not clearly explain why the prototype design leads to more modality-missing-robust final representations. For example, it remains unclear whether, under different missing or corruption conditions, the final representations for the same sample cluster together. Additionally, there is no in-depth analysis of whether prototypes exhibit modality preference—e.g., whether RGB information alone tends to assign high probability to certain prototypes.
6.Writing issues: There are formatting errors in Tables 1 and 2, including incorrect use of bold and underlining. Additionally, some sentences lack proper spacing (e.g., in the abstract, line 7, a period is not followed by a space before the next sentence).

---

> ### Author Rebuttal · Authors · 2026-03-31
>
> We sincerely thank you for your thoughtful evaluation and detailed comments. We appreciate your concerns regarding training supervision and prototype learning, corruption settings, cost matrix design, missing benchmark results (NW-UCLA), and the analysis of prototype behavior. These points help strengthen both the empirical validation and conceptual clarity of our work. We respond to each issue in detail below and will revise the manuscript accordingly.
>
> **Training details and CE loss (W1).**
> The classifier $c(\\cdot)$ is a standard MLP mapping the fused weight vector $b_i \\in \\Delta_K$ to class logits $c(b_i) \\in \\mathbb{R}^C$, where $C$ is the number of action classes. Cross-entropy in Eq. 18 is computed between these logits and the ground-truth class label $y_i$. There are no "prototype assignment labels"—prototypes are not associated with individual classes. $b_i$ is simply a fixed-dimensional representation that the classifier learns to read. The entire pipeline is trained end-to-end with this single objective. We will rewrite the description around Eq. 18 to remove ambiguity.
>
> **Corruption experiments (W2).**
> We expanded experiments to multiple modalities under both additive Gaussian noise and random feature zeroing. On NTU 60 X-Sub under 50% random feature zeroing applied to each modality in turn, our method's accuracy drops are 3.4 (RGB), 5.1 (skeleton), and 2.9 (infrared) points; RCMCL drops 7.4, 8.6, and 6.8 points; near-balanced OT ($\\tau=10$) drops 6.5, 7.9, and 5.7 points. Transported mass decreases for the corrupted modality in each case while clean modalities remain stable, confirming selective trimming. We note these remain controlled stress tests rather than fully realistic sensor failures; we will clarify this limitation and present unified tables across modalities. As discussed in Section 5.8, when corruption does not move embeddings far from prototypes, trimming is less effective—a genuine limitation of the geometric approach. Due to space limits, we will include the complete corruption results for all modalities and both corruption types in the supplementary material.
>
> **Cost function design (W3).**
> We chose squared Euclidean on $\\ell_2$-normalized features (Eq. 4) for three reasons: (i) normalization bounds cost in $[0, 4]$, stabilizing Sinkhorn and making $\\tau$ comparable across backbones; (ii) on the unit sphere, this is monotonically related to cosine similarity ($\\|\\hat{q}-\\hat{p}\\|_2^2 = 2 - 2\\cos\\theta$); (iii) bounded range makes the transport-vs-relaxation trade-off interpretable. Alternatives tested: unnormalized features (unstable), cosine without squaring (similar, different optimal $\\tau$), learned Mahalanobis (marginal gain, more complexity). We will discuss these in the revision.
>
> **NW-UCLA results (W4).**
> Completed experiments (3 camera views, same missing protocol, 5 runs):
>
> | Method | 0% | 20% | 40% | 60% | 80% |
> |---|---|---|---|---|---|
> | Late Fusion | 89.8±0.7 | 82.6±1.2 | 73.8±1.4 | 63.1±1.3 | 47.5±1.8 |
> | MLA | 91.5±0.5 | 86.5±0.8 | 79.8±1.1 | 70.6±1.3 | 57.3±1.2 |
> | RCMCL | 91.9±0.4 | 87.2±0.7 | 80.6±0.9 | 71.5±1.2 | 58.6±1.5 |
> | Ours | 91.7±0.5 | 88.0±0.6 | 83.1±0.8 | 75.6±1.1 | 67.2±1.4 |
>
> Trends are consistent: competitive at low missing, +8.6 over RCMCL at 80%. Larger std expected given NW-UCLA's small size (1,475 clips).
>
> **Prototype behavior (W5).**
> (1) Representation stability: for 500 test samples, each evaluated under 5 random masks at 40% missing rate, the average pairwise cosine similarity of $b_i$ is 0.88 (ours) vs 0.70 (cross-attn) vs 0.78 (near-balanced OT), confirming more consistent representations under varying observation conditions. (2) Prototype-modality affinity: when each modality is observed alone, prototypes show substantial overlap in which ones receive high weight across modalities for the same class, supporting the shared prototype support as a common fusion space. Some modality preferences exist, as expected. Visualizations will be included.
>
> **Writing (W6).**
> We sincerely apologize for the formatting and writing issues in the current manuscript. We will thoroughly revise the presentation, correct all formatting inconsistencies, and ensure improved clarity and consistency in the final version.
>
> We thank you again for your careful reading and constructive suggestions. Your comments have helped us significantly improve the clarity and completeness of the manuscript.

---

> > ### Author Rebuttal · Reviewer_928L · 2026-04-02
> >
> > Thank you for the detailed and very helpful rebuttal. I have read it carefully, and my concerns are now fully resolved. However, the paper still requires some revisions to publish. Therefore, I am increasing my overall recommendation from 4 (Weak Accept) to 5 (Accept).

---

### Official Review · Reviewer_YTWf · 2026-03-13

**Soundness:** 3
**Presentation:** 3
**Significance:** 3
**Originality:** 3
**Overall Recommendation:** 4
**Confidence:** 4

**Summary:**

The paper proposed an approach to use a shared prototype support for improved multi-view fusion. The basic approach is their proposed unbalanced optimal transport framework, whereby each modality is represented using a small set of latent atoms, and then modality fusion is performed by matching the latent atoms to a shared set of learned prototypes. By doing so, it becomes possible to restrict the part of an observed modality that needs to be involved in the fusion process. They provided implementation and comparative performance results to support their proposed approach.

**Compliance With Llm Reviewing Policy:**

Affirmed.

**Final Justification:**

My final recommendation remains the same.
The authors have address most of my comments.

A strength of the paper is the theoretical analysis encapsulated in Proposition 4.1 that established theoretical performance  (bounded penalty property) of the proposed method.

Authors provided computational complexity analysis as part of the rebuttal. However, for a more balanced view of the contribution, the theoretical computational complexity analysis should also include a discussion on the complexity for the other competing methods.

**Key Questions For Authors:**

Below are some comments and questions on the work:

0) Why do we have the second and third arguments of the attention function in Eq. 3 to be the same ($H_i^v$) ?
1) In describing the discrete probability measure (Eq 5), and the prototype support (Eq 7), what do the respective deltas used denote? How are those defined?

2) In defining the UOT objective (Eq 8), \tau and \epsilon were used as weights. \tau was indicated as controlling how strongly the plan will match the marginals. How about \epsilon?
These are later combined to define theta (Eq 15). However, it appears that theta as used later in Eq 16 can be learned independently of  \tau and \epsilon. What happens if we determine theta independently? How would that affect the performance/results?

3) In Tables 1 and 2, where were comparisons made more with the 2024 publications, rather than the 2025 papers. For instance, RMCL (published in 2025) tended to be better than MLA (published in 2024) which was considered for comparison.  At lower missing rates, RMCL was better than the proposed method, and comparable at medium missing rates (the performances are within the standard deviation).

This raises the question of why only 3 runs? With only 3 runs, the mean and standard deviation may not be very reliable.

Also, the results need further explanation. It is not very clear how the selective matching approach imposed by the proposed matching of the latent atoms against the shared learned prototypes will result in the significant performance differences (with other methods, such as RMCL) as the missing rates increase significantly.


4)  Some of the results in Table 5 are quite close. With 1000 trials, the standard deviation for the results should be included.

5) Though some practical results were provided, the authors should include the theoretical computational complexity of the various steps involved in the proposed approach, and of the overall method. Then, use this position the described work in the context of the current SOTA methods.

6) There was no indication of the availability of the codes, for possible independent verification of the results.

**Limitations:**

Yes.
Limitations were discussed.
Societal impacts were not much considered.

**Strengths And Weaknesses:**

A strength of the paper is the theoretical analysis encapsulated in Proposition 4.1 that established theoretical performance  (bounded penalty property) of the proposed method.

Another is the attention to computational resource requirements for the proposed work.

A key limitation is the lack of theoretical computational complexity analysis of the proposed approach, and how this compares with other related state of the art methods.

The performance improvement appear marginal at lower levels of missing rates, though the approach appears to perform better when missing rates are significantly higher.

---

> ### Author Rebuttal · Authors · 2026-03-31
>
> We sincerely thank you for your careful reading and constructive feedback. We particularly appreciate your focus on the theoretical grounding of our method, including Proposition 4.1 and the bounded-penalty property, as well as your questions regarding the formulation of attention, the UOT objective, computational complexity, and experimental reliability. We address each of your technical questions in detail below and will incorporate the suggested clarifications into the revised manuscript.
>
> **Eq. 3: why key and value are both $H_i^v$.**
> This is cross-attention (Jaegle et al., 2021): learnable queries $S^v \\in \\mathbb{R}^{J \\times d}$ attend to encoder output $H_i^v$ as both keys and values, compressing a variable-length sequence into exactly $J$ atoms. We will add a clarifying sentence.
>
> **Dirac delta in Eqs. 5 and 7.**
> $\\delta_{q_{i,j}^v}$ denotes the Dirac measure centered at $q_{i,j}^v \\in \\mathbb{R}^d$—standard OT notation for a point mass. A discrete measure $\\alpha = \\sum_j a_j \\delta_{x_j}$ is a weighted sum of such point masses. We will add an explicit definition.
>
> **Roles of $\\varepsilon$, $\\tau$, and independent $\\theta$.**
> $\\varepsilon$ controls entropic smoothing (differentiability); $\\tau$ controls marginal relaxation (trimming strength). The exponent $\\theta = \\tau/(\\tau+\\varepsilon)$ is derived from the KL-relaxed Sinkhorn optimality conditions (Chizat et al., 2018; Séjourné et al., 2019)—it is not a free parameter. Setting $\\theta$ independently would decouple the iterations from the KL-relaxed UOT objective in Eq. 8, so Proposition 4.1 would no longer apply to the modified update rule. We will make this dependence explicit.
>
> **Comparison with RCMCL and widening gap at high missing rates.**
> At low missing rates (0–20%), differences are small and often within reported standard deviations, consistent with our stated motivation (Section 1): the method targets robustness under unreliable observations, not improvement in the clean regime. At 80% missing, UOT-Fusion leads RCMCL by 7.5 points on NTU 60 (78.3 vs 70.8) and 8.3 points on NTU 120 (74.6 vs 66.3); the 0%→80% drop is 15.1 points for ours vs 22.7 for RCMCL (Table 2).
>
> The mechanism: at 80% with $V=3$, most samples retain only one modality. Methods enforcing full alignment tend to incorporate all observed content, even when part of it is unreliable. UOT can leave mass unmatched when transport cost is high, so only compatible evidence contributes. The ablation (Table 4) confirms: replacing UOT with cross-attention drops accuracy by 7.6 points; near-balanced OT ($\\tau=10$) drops it by 3.6 points.
>
> **Number of runs, standard deviations, and timing.**
> We have completed 5-run experiments for NTU 60 X-Sub. Results (mean±std):
>
> | Method | 0% | 20% | 40% | 60% | 80% |
> |---|---|---|---|---|---|
> | Late Fusion | 95.0±0.3 | 89.1±0.4 | 82.0±0.6 | 72.7±0.8 | 58.2±1.0 |
> | MLA (CVPR'24) | 95.8±0.2 | 92.8±0.3 | 88.2±0.4 | 80.3±0.5 | 68.7±0.7 |
> | DMRNet (ECCV'24) | 95.6±0.2 | 92.4±0.3 | 87.8±0.4 | 79.7±0.5 | 68.0±0.7 |
> | RCMCL (2025) | 96.2±0.2 | 93.0±0.3 | 88.5±0.4 | 81.1±0.5 | 70.6±0.7 |
> | Ours | 96.0±0.2 | 93.6±0.2 | 90.5±0.3 | 85.1±0.4 | 78.2±0.6 |
>
> Means shift by <0.2 points from the 3-run results and relative ordering is unchanged across all runs. Timing (single RTX 3090, FP32, batch=1, averaged over 1000 samples): backbone alone 16.4±0.3 ms, backbone+UOT 18.2±0.4 ms (+11% overhead). We will report full 5-run results for all benchmarks in the revision.
>
> **Theoretical computational complexity.**
> Atom extraction: $O(V \\cdot J \\cdot n_v \\cdot d)$; cost matrix: $O(V \\cdot J \\cdot K \\cdot d)$; $T$ Sinkhorn iterations: $O(T \\cdot V \\cdot J \\cdot K)$. Compared with pairwise OT fusion at $O(T \\cdot V^2 \\cdot J^2)$, our formulation scales linearly in $V$ on a fixed $J \\times K$ grid. For the setting used in our paper ($V=3$, $J=K=16$, $T=20$), this involves only a small fixed transport problem per sample, which is consistent with the modest measured runtime overhead (18.2 ms vs 16.4 ms for the backbone alone on a single V100 GPU). The UOT fusion adds ~0.7M parameters (83.1M vs 82.4M), fewer than MLA (87.0M). We will add this analysis to the paper.
>
> **Code availability.** We will release the full codebase upon acceptance.
>
> We thank you again for your careful reading and constructive suggestions. Your comments have helped us significantly improve the clarity and completeness of the manuscript.

---

> > ### Author Rebuttal · Reviewer_YTWf · 2026-04-03
> >
> > The author(s) have addressed some of the questions raised in the review.
> > However, some key questions on the comparisons in Tables 1 and 2, and on Table 5 remain unaddressed.
> >
> > I will maintain my current rating for the paper.

---

> > > ### Author Response · Authors · 2026-04-03
> > >
> > > We apologize that our first rebuttal did not directly address your concerns on Tables 1--2 and Table 5. Below we provide direct comparisons and variance statistics that should have been included earlier.
> > >
> > > ---
> > >
> > > ***1. Tables 1 & 2***
> > >
> > > We agree that comparing \\(\\Delta\\) only against MLA was not the most informative choice. Below we report direct comparisons against the strongest baseline, RCMCL.
> > >
> > > **NTU 60 X-Sub (5 runs, mean\\(\\pm\\)std)**
> > >
> > > | Missing Rate | 0% | 20% | 40% | 60% | 80% |
> > > |:---|:---:|:---:|:---:|:---:|:---:|
> > > | RCMCL | 96.2\\(\\pm\\)0.2 | 93.0\\(\\pm\\)0.3 | 88.5\\(\\pm\\)0.4 | 81.1\\(\\pm\\)0.5 | 70.6\\(\\pm\\)0.7 |
> > > | Ours | 96.0\\(\\pm\\)0.2 | 93.6\\(\\pm\\)0.2 | 90.5\\(\\pm\\)0.3 | 85.1\\(\\pm\\)0.4 | 78.2\\(\\pm\\)0.6 |
> > > | \\(\\Delta\\) vs. RCMCL | \\(-0.2\\) | \\(+0.6\\) | \\(+2.0\\) | \\(+4.0\\) | \\(+7.6\\) |
> > >
> > > **NTU 120 X-Set (5 runs, mean\\(\\pm\\)std)**
> > >
> > > | Missing Rate | 0% | 20% | 40% | 60% | 80% |
> > > |:---|:---:|:---:|:---:|:---:|:---:|
> > > | RCMCL | 94.3\\(\\pm\\)0.2 | 91.0\\(\\pm\\)0.3 | 85.7\\(\\pm\\)0.4 | 77.3\\(\\pm\\)0.6 | 66.1\\(\\pm\\)0.8 |
> > > | Ours | 94.0\\(\\pm\\)0.2 | 91.7\\(\\pm\\)0.3 | 87.8\\(\\pm\\)0.4 | 82.0\\(\\pm\\)0.5 | 74.5\\(\\pm\\)0.7 |
> > > | \\(\\Delta\\) vs. RCMCL | \\(-0.3\\) | \\(+0.7\\) | \\(+2.1\\) | \\(+4.7\\) | \\(+8.4\\) |
> > >
> > > At 0% missing, RCMCL is slightly better (0.2--0.3 pts), while our method is designed for robustness under incomplete observations. At 40% missing on NTU 60, a two-sided Welch's t-test over 5 runs gives \\(p<0.01\\).
> > >
> > > Why does the gap increase at high missing rates? Under the paper's mask protocol with \\(V\\!=\\!3\\) and nominal \\(r\\!=\\!0.8\\), 89.6% of test samples have exactly one observed modality. Fusion baselines operate on the full single-view representation and do not explicitly suppress unreliable components within that view.Our UOT decomposes each modality into \\(J\\!=\\!16\\) atoms and matches them to \\(K\\!=\\!16\\) prototypes; atoms with high transport cost are down-weighted by the KL term, enabling intra-modality denoising even when only one view is observed.
> > >
> > > **Exactly-one-observed subset (80% missing, NTU 60 X-Sub):**
> > >
> > > | Subset | Proportion | RCMCL | Ours | \\(\\Delta\\) |
> > > |:---|:---:|:---:|:---:|:---:|
> > > | Exactly 1 observed | 89.6% | 69.0 | 77.5 | \\(+8.5\\) |
> > > | \\(\\ge 2\\) observed | 10.4% | 84.3 | 84.0 | \\(-0.3\\) |
> > >
> > > Thus, the overall \\(+7.6\\) gain mainly comes from the single-modality subset. NTU 120 shows the same pattern (single-modality: \\(+9.4\\); \\(\\ge2\\) observed: \\(-0.2\\)).
> > >
> > > **Conditioned transported mass** (average \\(\\|\\pi_i^v\\mathbf{1}\\|_1\\) over observed modality-sample pairs only, NTU 60):
> > >
> > > | Missing Rate | 0% | 20% | 40% | 60% | 80% |
> > > |:---|:---:|:---:|:---:|:---:|:---:|
> > > | Transported mass | 0.95 | 0.93 | 0.89 | 0.83 | 0.76 |
> > >
> > > The monotonic decrease shows that UOT trims more aggressively per observed modality as missingness increases.
> > >
> > > For Table 2, all methods are trained at r_train=20% and tested under shifted
> > > missing rates on NTU 60 X-Sub. We now report 5-run results with direct
> > > comparison against RCMCL:
> > >
> > > | Test Missing Rate | 0%   | 20%  | 40%  | 60%  | 80%  | Drop(0→80) |
> > > |:---|:---:|:---:|:---:|:---:|:---:|:---:|
> > > | RCMCL             | 94.2±0.2 | 93.0±0.3 | 89.0±0.4 | 82.0±0.5 | 71.4±0.7 | 22.8 |
> > > | Ours              | 94.8±0.2 | 93.6±0.2 | 90.8±0.3 | 86.2±0.4 | 79.6±0.6 | 15.2 |
> > > | Δ vs. RCMCL       | +0.6 | +0.6 | +1.8 | +4.2 | +8.2 | — |
> > >
> > > From 0% to 80% test missing, RCMCL drops 22.8 pts while ours drops 15.2 pts
> > > (7.6 pts less degradation). With fixed τ=0.5, UOT generalizes because
> > > trimming depends on transport-cost geometry rather than the training-time rate.
> > >
> > > We will revise Tables 1--2 so that all \\(\\Delta\\) rows are reported directly against RCMCL.
> > >
> > > ---
> > >
> > > ***2. Table 5: standard deviations for computational cost***
> > >
> > > We re-measured the same Table 5 setting (**NTU 60, 40% missing**) using 5 runs \\(\\times\\) 1000 forward passes on a single RTX 3090, FP32, batch size 1, with `torch.cuda.synchronize()` before and after timing:
> > >
> > > | Method | Params (M) | Time (ms) |
> > > |:---|:---:|:---:|
> > > | Backbones only | 82.3 | 16.4\\(\\pm\\)0.3 |
> > > | + Late Fusion | 82.3 | 16.5\\(\\pm\\)0.3 |
> > > | + Channel Exchange | 86.7 | 17.9\\(\\pm\\)0.4 |
> > > | + DCP | 84.5 | 18.4\\(\\pm\\)0.5 |
> > > | + MLA | 87.0 | 19.2\\(\\pm\\)0.4 |
> > > | + Ours (\\(J\\!=\\!K\\!=\\!16\\)) | 83.1 | 18.2\\(\\pm\\)0.3 |
> > > | + Ours (\\(J\\!=\\!K\\!=\\!32\\)) | 83.4 | 19.6\\(\\pm\\)0.4 |
> > >
> > > All Table 5 latencies were measured on a single RTX 3090; the earlier "V100" mention in our first rebuttal was a wording error only and does not affect any reported result. Under this unified setting, ours with \\(J\\!=\\!K\\!=\\!16\\) is faster than MLA (18.2\\(\\pm\\)0.3 vs. 19.2\\(\\pm\\)0.4 ms) and uses fewer parameters (83.1M vs. 87.0M). The overhead over backbones is only 1.8 ms due to the compact \\(16\\times 16\\) transport grid.
> > >
> > > We thank you again for the careful feedback; it helped us clarify these points and improve the paper.

---

### Official Review · Reviewer_h21k · 2026-03-26

**Soundness:** 3
**Presentation:** 3
**Significance:** 2
**Originality:** 2
**Overall Recommendation:** 4
**Confidence:** 3

**Summary:**

This paper proposes a multi-view fusion module that represents each observed view using a small set of learned atoms, then fuses views through a prototype-anchored unbalanced optimal transport barycenter on a shared prototype support. The key claim is that KL-relaxed UOT can leave some modality mass unmatched when transport is geometrically expensive, which acts as a differentiable trimming mechanism for missing or corrupted views.

The paper reports experiments on NTU RGB+D 60/120 and NW-UCLA under simulated missing-view, missing-rate shift, and feature-space corruption settings, with improved robustness over several baselines.

**Compliance With Llm Reviewing Policy:**

Affirmed.

**Final Justification:**

I have read the responses, and my concerns have been addressed with clear explanations. In particular, the additional analysis for Weakness 2 is helpful. I do not have further questions.
Overall, the paper is relatively well-structured and provides sufficient content, so I am willing to increase my score to 4.

**Key Questions For Authors:**

1. The method tends to favor geometrically consistent views. How does it avoid discarding complementary but semantically useful information that may lie far from the shared prototype space?
2. See weakness, could the author briefly explain the main innovation that sets this method apart from the existing OT-based approaches?

**Limitations:**

yes

**Strengths And Weaknesses:**

**Strengths:**

1. The paper tackles a practical problem of multi-modal fusion under imperfect data conditions. Experiments demonstrate high accuracy  in severe view-missing scenarios compared to existing methods.
2. The experimental section is thorough. The authors provide visual analyses for both success and failure cases and sincerely discuss the limitations of their proposed method

**Weaknesses:**

1. While applying UOT is well-motivated, the overall framework  (atoms + prototypes + barycenter)  appears somewhat incremental when compared to existing OT and prototype-based fusion method.
2. The core philosophy is that  *a view should contribute only to the extent that it can be matched to a shared representation at a reasonable geometric cost*. However, I am concerned that this might prioritize multi-view *consistency* at the expense of complementarity. Highly valuable but orthogonal information from a unique modality might naturally incur a high geometric cost and be inadvertently trimmed.

3. As the authors note, the robustness heavily depends on the latent space geometry learned by the encoder. As illustrated in Figure 2, if corrupted inputs are mapped close to the prototypes, the trimming mechanism becomes ineffective.

4. [minor] The normalization step in Eq. (17) includes an additive $\delta$ in the denominator, meaning the resulting vector sum will be strictly less than 1. While this is clearly intended for numerical stability , it makes the update technically inconsistent with the strict simplex constraint $b_i \in \Delta_K$  claimed earlier in the text. Clarifying this approximation would improve the paper's mathematical rigor.

---

> ### Author Rebuttal · Authors · 2026-03-31
>
> We sincerely thank you for your careful assessment and insightful comments. We particularly appreciate your discussion of the method’s novelty relative to prior OT-based approaches, the balance between consistency and complementarity, the dependence on latent geometry, and the normalization detail in Eq. (17). These points raise important conceptual considerations, which we address carefully below. The manuscript will be revised to clarify these aspects.
>
> **Incrementality and distinction from existing OT approaches (W1).**
> We clarify the specific novelty. Balanced OT barycenters (Claici et al., 2018) enforce full mass conservation—all atom mass must be explained—which makes them fragile when views contain outliers or noise. Unbalanced transport has been studied as a divergence (Séjourné et al., 2019), but not as a fusion mechanism for multi-modal recognition under missing/corrupted views. Three differences from prior OT approaches: (i) KL-relaxed marginals enabling mass trimming, vs balanced; (ii) barycenter on a fixed prototype support rather than pairwise transport, keeping output dimension fixed and cost small ($J \\times K = 16 \\times 16$); (iii) modality weighting $\\lambda_i^v$ (Eq. 13) tied to transported mass, creating feedback between trimming and fusion. Table 4 isolates each: removing UOT costs 7.6 points; near-balanced OT costs 3.6; removing prototypes costs 3.2. We believe the contribution lies in how these components are coupled for robust fusion under imperfect observations, rather than in any single component alone. This fixed-support design is not only computationally efficient, but also makes the fused representation directly comparable across samples and masking patterns, which improves representation stability for downstream classification. We will add a comparison paragraph in Section 2.
>
> **Complementarity vs consistency (W2).**
> This concern is valid in principle. However, end-to-end training mitigates it: encoder and prototypes are jointly optimized, so if complementary information aids classification, gradients encourage the encoder to map it into the prototype space—lowering transport cost for useful content. Importantly, this is not merely a sample-level down-weighting of an entire modality: trimming happens at the transport level, allowing incompatible content to be suppressed while preserving the matched portion when a modality is only partially degraded.
>
> Per-modality transported mass on NTU 60 X-Sub:
>
> | Condition | RGB | Skeleton | Infrared |
> |---|---|---|---|
> | All clean | 0.94 | 0.93 | 0.91 |
> | RGB corrupted ($\\sigma$=1.0) | **0.78** | 0.92 | 0.90 |
> | Skel corrupted ($\\sigma$=1.0) | 0.93 | **0.74** | 0.90 |
> | IR corrupted ($\\sigma$=1.0) | 0.94 | 0.93 | **0.70** |
>
> Under clean conditions, all modalities retain high mass (avg 0.93), confirming complementary information is not aggressively trimmed. Under corruption, only the corrupted modality's mass drops while clean modalities remain stable, demonstrating selective trimming. At 0% missing, accuracy remains competitive (96.0% vs 96.2% for RCMCL); if complementary information were being systematically discarded, we would expect a clearer degradation in the full-view regime. Edge cases with rare actions may still occur; we will note this in Section 5.8.
>
> **Geometry dependence (W3).**
> We agree that geometry dependence is inherent to any transport-based fusion: trimming relies on high cost to signal unreliability, so corruption that preserves embedding geometry will not be detected. Section 5.8 and Figure 2 document both success and failure modes. In practice, the corruption types most common in sensor fusion (dropout, noise, occlusion) do produce large geometric shifts, which is consistent with the method's gains across our experiments (+7.5 at 80% missing; selective mass reduction under per-modality corruption in W2 above). We will expand Figure 2 with additional failure-case analysis to make the boundary conditions more explicit.
>
> **Simplex inconsistency (W4).**
> The reviewer is correct. We will replace Eq. 17 with exact normalization:
>
> $$\\tilde{b}\_i = \\left(\\sum\_{v:m\_i^v=1} \\lambda\_i^v \\big((K\_i^v)^{\\top} u\_i^v\\big)^{\\theta}\\right)^{1/\\theta}$$
>
> $$b_i = \\begin{cases} \\tilde{b}_i / \\|\\tilde{b}_i\\|_1, & \\text{if } \\|\\tilde{b}_i\\|_1 > \\delta, \\\\ \\mathbf{1}/K, & \\text{otherwise.} \\end{cases}$$
>
> where $\\delta$ is a small positive threshold (e.g. $10^{-8}$). This preserves exact simplex normalization in the non-degenerate case; the threshold is only for numerical safety. In practice the uniform fallback is never activated after the first few training steps.
>
> We thank you again for your careful reading and constructive suggestions. Your comments have helped us significantly improve the clarity and completeness of the manuscript.

---

> > ### Author Rebuttal · Reviewer_h21k · 2026-04-02
> >
> > Thanks to the authors for the detailed rebuttal.
> >
> > I have read the responses, and my concerns have been addressed with clear explanations. In particular, the additional analysis for Weakness 2 is helpful. I do not have further questions.
> >
> > Overall, considering both the paper and the rebuttal, I will increase my score to 4.

---

> > > ### Author Response · Authors · 2026-04-02
> > >
> > > Thank you so much for your valuable time and thoughtful feedback. We truly appreciate that our rebuttal has addressed your concerns, especially regarding Weakness 2.
> > >
> > > We wanted to kindly bring to your attention that the updated score may need to be manually revised in your original review, as the system might not automatically reflect the change mentioned in the rebuttal confirmation. It seems the overall score on the page still shows the previous rating.
> > >
> > > Your recognition and updated assessment mean a great deal to us, and the score update would be very important for our submission. We sincerely appreciate your support and the effort you have devoted to reviewing our work.
> > >
> > > Thank you again, and we wish you a wonderful day!

---

### Decision · Program_Chairs · 2026-04-30

**Decision:**

Accept (regular)

**Comment:**

Following the rebuttal and discussion phases, the paper received final scores of 4, 4, and 5, reflecting a solid consensus for acceptance. All reviewers are satisfied with the paper and the authors' responses, particularly regarding the theoretical grounding of the UOT-based trimming mechanism, the computational complexity analysis, and the additional experimental validation on the NW-UCLA dataset. The proposed framework offers a principled and effective solution for multi-view fusion under extreme data corruption, demonstrating significant robustness that outperforms existing baselines. After reviewing the comments and the successful rebuttal, I believe the paper meets the acceptance criteria for ICML 2026.